# Identification of Typical Ecosystem Types by Integrating Active and Passive Time Series Data of the Guangdong–Hong Kong–Macao Greater Bay Area, China

**DOI:** 10.3390/ijerph192215108

**Published:** 2022-11-16

**Authors:** Changlong Li, Yan Wang, Zhihai Gao, Bin Sun, He Xing, Yu Zang

**Affiliations:** 1School of Information Technology and Engineering, Guangzhou College of Commerce, Guangzhou 511363, China; 2Institute of Forest Resource Information Techniques, Chinese Academy of Forestry, Beijing 100091, China; 3Key Laboratory of Forestry Remote Sensing and Information System, NFGA, Beijing 100091, China; 4Shandong Geographical Institute of Land Spatial Data and Remote Sensing Technology, Jinan 250002, China

**Keywords:** Guangdong–Hong Kong–Macao greater bay area (GBA), typical ecosystem types, integrating active and passive data, time series data

## Abstract

The identification of ecosystem types is important in ecological environmental assessment. However, due to cloud and rain and complex land cover characteristics, commonly used ecosystem identification methods have always lacked accuracy in subtropical urban agglomerations. In this study, China’s Guangdong–Hong Kong–Macao Greater Bay Area (GBA) was taken as a study area, and the Sentinel-1 and Sentinel-2 data were used as the fusion of active and passive remote sensing data with time series data to distinguish typical ecosystem types in subtropical urban agglomerations. Our results showed the following: (1) The importance of different features varies widely in different types of ecosystems. For grassland and arable land, two specific texture features (VV_dvar and VH_diss) are most important; in forest and mangrove areas, synthetic-aperture radar (SAR) data for the months of October and September are most important. (2) The use of active time series remote sensing data can significantly improve the classification accuracy by 3.33%, while passive time series remote sensing data improves by 4.76%. When they are integrated, accuracy is further improved, reaching a level of 84.29%. (3) Time series passive data (NDVI) serve best to distinguish grassland from arable land, while time series active data (SAR data) are best able to distinguish mangrove from forest. The integration of active and passive time series data also improves precision in distinguishing vegetation ecosystem types, such as forest, mangrove, arable land, and, especially, grassland, where the accuracy increased by 21.88%. By obtaining real-time and more accurate land cover type change information, this study could better serve regional change detection and ecosystem service function assessment at different scales, thereby supporting decision makers in urban agglomerations.

## 1. Introduction

With the continuous deepening of reform and opening up and the strong support of national policies, the Guangdong–Hong Kong–Macao Greater Bay Area (GBA) has become an important foundation for economic and social development at a regional level in China [1]. At the same time, this urban agglomeration expansion has been accompanied by changes in urban population and landscape patterns, resulting in intensified changes in land cover and land use [2,3]. In particular, many natural ecosystems (such as forest ecosystems, grassland ecosystems, and so on) are occupied or gradually fragmented. In this way, on the one hand, the habitats of wild animals and plants are destroyed and biodiversity is drastically reduced [4,5]. On the other hand, the destruction of the landscape pattern reduces or loses many service functions of the ecosystem [6,7]. All of these have greatly hindered the further healthy and harmonious development of the urban agglomeration. Therefore, it is important to establish a quicker and more accurate method of identifying land cover types in the complex ecosystems of GBA and subtropical urban agglomeration.

At present, remote sensing technology is widely used to identify land cover and land use in various ecosystems, such as surface waters [8], wetlands [9], savannahs [10], grasslands [11], shrublands [12], forests [13], forest-agriculture mosaics [14], oasis agricultural crops [15], temperate and Mediterranean agricultural crops [16], urban areas [17], and so on. Meanwhile, the identification of typical ecosystem types based on various models has been applied in much research worldwide, such as India and other tropical and subtropical regions [18,19,20]. For GBA, many scholars have carried out in-depth research on the classification of land cover types in the GBA [21,22]. They mainly focus on computer-based deep learning techniques for a variety of parametric, non-parametric, supervised, and unsupervised classification methods, including support vector machines, neural networks, maximum likelihood classification, minimum distance classification, decision tree classification, spectral mixing analysis, spectral information divergence, and the spectral angle mapper [23]. However, remote sensing image classification is a complex process, and no single classifier can achieve high-precision classification and extraction for all land cover types in all regions [24]. For this reason, some scholars now use multi-classifier combination methods for the classification of ground objects, using a form of weighted or unweighted voting [25,26].

However, it is difficult to accurately distinguish different vegetation types, such as forests, grasslands, and cultivated land, using single-temporal remote sensing data in subtropical urban agglomerations because its effectiveness is impaired by factors such as long periods of cloudy weather, complex ecosystem types, and shortened change cycles [27]. Typically, in these settings, single-temporal remote sensing data fail to fully describe land cover characteristics and changes [28]. In response, many scholars have sought to improve classification accuracy with respect to different land cover types by extracting more detailed features, such as texture, or by applying more advanced classification methods, such as deep learning [29,30,31]. Texture analysis offers a means of extracting the spatial features of the image itself and digging deep into the image information. Due to the particular spatial distributions of different land cover types, the use of texture features in remote sensing image classification can greatly improve the accuracy of results [32]. However, because of limitations in the data themselves, such techniques still cannot meet the needs of land use professionals involved in actual production processes or in-depth scientific research.

Synthetic-aperture radar (SAR) can identify and monitor land cover types throughout the day and in all weathers because it can exclude the influence of weather conditions such as clouds and rain [33]. In recent years, especially since the launch of Sentinel-1, the application of SAR data has expanded in projects such as the mapping of agricultural land cover in the Camargue region of France [34], the mapping of forest land cover types, such as temperate broad-leaved forest, boreal coniferous forest, and montane forest [35], and the mapping of vegetation types in cities such as Lyon, Cologne, Prague, France, Germany, and the Czech Republic, respectively [36]. The comprehensive application of optical data and SAR data has also improved the accuracy of classification results [37]. Tavares et al. [38] integrated Sentinel-1 and Sentinel-2 data to classify land cover/land use types in the tropical Amazon region of Brazil and found a 2% improvement in classification accuracy, compared with a single optical data source. Zhang et al. [39] integrated multi-band optical data from Landsat and SAR data from the ALOS-2 satellite to identify and classify land types in cloudy mountainous areas and also obtained significantly improved accuracy overall.

Additionally, the use of time series optical data is more likely to accurately distinguish different vegetation types, such as forests, grasslands, and cultivated land, using single-temporal remote sensing data [40]. Time series data for a whole year is able to capture seasonal characteristics of vegetation features, and because different vegetation features exhibit different characteristics, time series data can more accurately distinguish vegetation features [41]. Persson et al. [42] used Sentinel-2 data across the four seasons of the year to classify forest tree species in central Sweden and obtained an overall accuracy of 88.2%, far exceeding the classification accuracy of single-temporal data. Liu et al. [43] used multi-seasonal data from the RedEdge-MX sensor to classify and identify urban tree species and found that results were significantly affected by imaging time. When multi-temporal data were incorporated, classification accuracy rose to a level of 92.16%. Van et al. [44] sought to distinguish between communities of wetland and dryland vegetation in the subtropical coastal region of South Africa. They found that four-season methods produced more accurate classification results than single-season methods and further demonstrated that multi-seasonal imagery improves the identification of separate communities in a region dominated by evergreen species.

The use of integrated optical data and SAR data is now well established, as is the use of multi-temporal image data to identify vegetation and other land cover types [45]. However, the integration of active and passive time series SAR data remains rare, and time series data, such as the normalized difference vegetation index (NDVI), have been used to identify land cover types in urban agglomerations, such as the GBA, on a monthly basis only [46]. Therefore, in this study, the Sentinel-1 and Sentinel-2 data was used as the time series active and passive data sources, and the simple non-iterative clustering (SNIC) segmentation algorithm and the random forest (RF) classification algorithm were used as the study methods to identify and analyze the complex ecosystem types of GBA urban agglomeration. It is expected that the advantages and disadvantages of active and passive remote sensing data can be obtained in GBA complex ecosystem type identification, and then a faster and more accurate method should be developed to identify land cover types in complex ecosystems of subtropical urban agglomerations. These data could be incorporated into the logical steps and could play different roles in project cycles [47,48]. For example, the change information of land cover type can enrich the basic ecological resources possessed by project management in the context analysis and help to formulate and put into action suitable solutions and decisions in planning and input phases. Finally, the real-time dynamic change of land cover type can not only promote the processing of project management, but also monitor its implementation effect. Therefore, the results provide important information support for decision-makers in urban agglomerations and researchers of ecological effects.

## 2. Materials and Methods

### 2.1. Study Area

The GBA is a world-class bay area connected by land and sea, mountains and rivers. It is an important space carrier for China to build a world-class city cluster and participate in global competition [49], it has an important strategic position in the national development pattern, and it is the fourth largest bay area in the world, after the San Francisco Bay Area, the San Francisco Bay Area in America, and the Tokyo Bay Area in Japan [50]. The geographical location of GBA is shown in Figure 1. The urban agglomeration is located in the coastal area of southwest China, from 21.59–24.37° N and 113.73–115.41° E. The total area is 56,000 km^2^, in a region with a subtropical monsoon climate, and the forest area is 27,481 km^2^, with forest coverage of approximately 42%. The study area includes four major river systems, many rich wetland ecosystems, including mangrove wetlands, and a coastline of more than 1500 km. In the course of the reform and opening-up of China, urban construction in the GBA has rapidly developed. The total urban area has expanded from 2805.48 km^2^ in 1978 to 7294.66 km^2^ in 2018, representing an increase of 160% in 40 years [51,52]. From 2010 to 2020, the population concentration of major urban agglomerations in China increased rapidly, and the population of GBA has increased by 35% to 86.1719 million. In 2021, the GDP of the GBA was about 12.6 trillion yuan, an increase of 2.4 trillion yuan in 5 years compared with 2017, reaching the level of developed countries in the world [53,54]. In summary, the GBA is an urban agglomeration with various complex ecosystems, including cities, forests, wetlands, oceans, rivers, and farmland. At the same time, with the ongoing construction of cities, its types of land cover are changing rapidly.

### 2.2. Data Acquisition and Preprocessing

For remote sensing data, the Sentinel-1 SAR Ground Range Detected (GRD) data and Sentinel-2 Multispectral Instrument (MSI) data were used. Each scene of Sentinel-1 GRD data was pre-processed with Sentinel-1 Toolbox using the following steps: thermal noise removal; radiometric calibration; terrain correction using SRTM 30, or ASTER DEM for areas greater than 60 degrees latitude, where SRTM is not available. The final terrain-corrected values are converted to decibels via log scaling (10*log10(x)). GRD and MSI products could also be fusion through this toolbox. The polarization modes of the Sentinel-1 GRD data were VV and VH, the spatial resolution was 10 m, and the imaging time was 2020. We divided the whole year into 24 time spans, half a month being one time span. In each time span, all remote sensing images were de-clouded according to each pixel, and then the mean value was calculated.

L1C tiles were used in the Sentinel-2 MSI data, and the texture features were extracted based on data from July 2020. The spatial resolution of Sentinel-2 data was 10 m, and the imaging time was 2020. Images of different spatial resolutions were unified to 10m resolution by nearest neighbor resampling. The multi-band image data were obtained based on the July image after cloud removal and were used to select an optimal value according to the pixel. The time series NDVI data were obtained by calculating the optimal value of the image after the first and second halves (1H, 2H) of each month (a total of 24 scenes).

The training sample data were collected by manual interpretation based on sub-meter resolution data from Google Earth, with a total of 190 sample points. The high-resolution image data on Google Earth was spliced from data acquired over many years, and the time span was from 2012 to 2021. When selecting sample data in this study, only high-resolution images formed in 2019, 2020, and 2021 were used. The data were obtained to ensure that the selected sample points could represent the land cover types when the sentinel data was imaged. The validation data were randomly generated at 120 sample points and then manually interpreted based on Google Earth. The specific experimental steps were: randomly generate 120 sample points on the ArcGIS software (Environmental Systems Research Institute, Inc. (Esri), Redlands, CA, USA) and then overlap all the sample points on Google Earth. Since the highest spatial resolution of Google Earth reaches 0.3 m, the land cover types can be clearly identified from the images. Therefore, the types to which all sample points belong are manually determined, and finally the validation data are obtained. Figure 1 shows the distribution. The collection of remote sensing data, the preprocessing, and implementation of classification methods and the validation of results were all based on the Google Earth Engine (GEE) platform.

### 2.3. Classification System and Features Set

Based on the characteristics of the study area, and on our ability to extract land cover information from Sentinel remote sensing images, land cover types were divided into six categories: forest, grass, arable land, impermeable layer, mangrove, and water. Table 1 gives definitions of each type.

Based on geographical characteristics, vegetation growth status, and phenological characteristics, the data were selected—including multi-band optical images—from Sentinel-2 and used to calculate characteristic indices (NDVI, EVI, NDWI, RGRI, NDBI). From Sentinel-1, time series SAR data were obtained based on VV and VH polarization and multi-dimensional texture features based on SAR data for the vegetation growth period (July). The feature set is shown in Table 2, and the calculation formulas of the index features are as follows:(1)$$\mathrm{NDVI}\;=\frac{\mathrm{NIR}-\mathrm{Red}}{\mathrm{NIR}+\mathrm{Red}}$$
(2)$$\mathrm{EVI}=2.5\left(\mathrm{NIR}-\mathrm{Red}\right)/\left(\mathrm{Nir}+C_{1}\mathrm{Red}-C_{2}\mathrm{Blue}+L\right)$$
(3)$$\mathrm{NDWI}\;=\frac{\mathrm{Green}-\mathrm{NIR}}{\mathrm{Green}+\mathrm{NIR}}$$
(4)$$\mathrm{RGRI}\;=\frac{\mathrm{Red}}{\mathrm{Green}}$$
(5)$$\mathrm{NDBI}\;=\frac{\mathrm{MIR}-\mathrm{NIR}}{\mathrm{MIR}+\mathrm{NIR}}$$
where Blue is the blue band, Green is the green band, Red is the red band, NIR is the near-infrared band, MIR is the mid-infrared band, and C_1_, C_2_, and L are the adjustment coefficients.

Due to the large amount of feature data, variable importance measures (VIMs) were used based on the Gini index (GI) from the random forest algorithm to select the features [55]. Assuming that there are *j* features *X*_1_, *X*_2_, …, *X_j_*, *i* is the number of decision trees and *C* is the number of categories, the GI of each feature can be expressed as:(6)$${\mathrm{GI}}_{q}^{\left(i\right)}=\overset{\left|C\right|}{\underset{c=1}{\sum }}\underset{c^{\prime }\;\neq \;c}{\sum }p_{qc}^{\left(i\right)}p_{qc^{\prime }}^{\left(i\right)}=1-\overset{\left|C\right|}{\underset{c=1}{\sum }}{\left(p_{qc}^{\left(i\right)}\right)}^{2}$$
where *p_qc_* represents the proportion of category *c* in node *q*. VIM can then be expressed as:(7)$${\mathrm{VIM}}_{jq}^{\left(Gini\right)\left(i\right)}={\mathrm{GI}}_{q}^{\left(i\right)}-{\mathrm{GI}}_{r}^{\left(i\right)}-{\mathrm{GI}}_{l}^{\left(i\right)}$$
where ${\mathrm{GI}}_{r}^{\left(i\right)}$ and ${\mathrm{GI}}_{l}^{\left(i\right)}$, respectively, represent the GI of the two new nodes after the branch. The total VIM of feature set *X_j_* in *i* can now be expressed as:(8)$${\mathrm{VIM}}_{j}^{\left(Gini\right)}=\overset{I}{\underset{i=1}{\sum }}{\mathrm{VIM}}_{j}^{\left(Gini\right)\left(i\right)}$$

Finally, VIM was normalized.

### 2.4. Study Methods

#### 2.4.1. SNIC Segmentation Algorithm

The algorithm determines optimal distance after selection of a seed point [56]. Assuming that *N* pixels are pre-segmented into *K* superpixels, the size of each superpixel is approximately *N*/*K* and the distance between adjacent seed points is approximately:(9)$$S=\sqrt{N/K}$$

The optimal distance was now determined. Each pixel has two parts: spatial distance and color distance. For each pixel *j*, its spatial distance is ($x_{j},y_{j}$), and its color distance is ($l_{j},a_{j},b_{j}$). Its distance from the *k*th seed point is therefore:(10)$$D_{s}=\sqrt{{\left(x_{j}-x_{k}\right)}^{2}+{\left(y_{j}-y_{k}\right)}^{2}}$$
(11)$$D_{c}=\sqrt{{\left(l_{j}-l_{k}\right)}^{2}+{\left(a_{j}-a_{k}\right)}^{2}+{\left(b_{j}-b_{k}\right)}^{2}}$$
(12)$$D=\sqrt{\frac{D_{s}^{2}}{s}+\frac{D_{c}^{2}}{m}}$$
where *D_s_* represents the color distance, *D_c_* represents the spatial distance, *D* represents the intra-class distance, and *s* and *m* represent the normalization parameters of the color distance and spatial distance, respectively.

Finally, the pixel clustering was implemented based on the priority queue algorithm. This can be optimized for multiple iterations into one iteration, which greatly improves the running efficiency. The algorithm establishes *k* elements based on *k* seed points, and the formula is as follows:(13)$$e_{i}=(x_{i},y_{i}),(l_{i},a_{i},b_{i}),k,D_{i,k}$$
where *k* is the super-pixel label and $D_{i,k}$ represents the distance from pixel *i* to the *k*th seed point. The queue returns the element *e_i_* with the smallest distance $D_{i,k}$ to the *k*th seed point each time.

#### 2.4.2. RF Classification Algorithm

We chose the decision tree classification algorithm mainly because the algorithm can not only perform deep learning based on image features, but also integrate multiple classifiers to achieve the optimal value of the algorithm. More importantly, the algorithm can evaluate the importance of all participating image features. Analysis can better help us to understand the impact of active and passive remote sensing data on GBA classification [57]. Each decision tree is a classifier, which randomly selects samples (usually 2/3 of the total number of training samples) and features, improving the fitting ability of the algorithm and reducing the effects of abnormal samples. The sampling method uses a bootstrap algorithm. After each selection of samples and features, multiple decision tree models were established to obtain multiple classification results. A final classification was then determined by voting. For the evaluation of the model, the out-of-bag error was used, that is, the ratio of the number of misclassifications to the total number of samples. This ratio expresses the misclassification rate of the random forest [58].

#### 2.4.3. Accuracy Evaluation

The confusion matrix was used to evaluate classification accuracy, including producer accuracy, user accuracy, overall accuracy, and Kappa coefficients.

## 3. Results

### 3.1. Single-Temporal Remote Sensing Image Classification

Our feature analysis of single-temporal remote sensing images (Figure 2) showed that the importance scores for sum average in VH polarization image (VH_savg), sum average in VV polarization image (VV_savg), dissimilarity in VV polarization image (VV_diss), and difference variance in VH polarization image (VH_dvar) were 31.19%, 28.02%, 19.44%, and 17.73%, respectively, forming the top four rankings of all texture features. Therefore, these four texture features were selected and used together with the optical remote sensing images for the classification of single-temporal remote sensing images.

The single-temporal image classification results are shown in Figure 3 and Table 3. The two texture features (VV_dvar and VH_diss) have the highest importance scores and the greatest effect on the classification results. The effects of EVI and B2 are similar and rank third and fourth, respectively, in the feature importance scores. These classification results show arable lands and grasslands seriously misclassified in the western part of the study area. The confusion matrix also shows lower classification accuracy with respect to arable, grass, and mangrove areas, with an overall figure of just 77.62%. In addition, the producer accuracy of the water is 100% and the user accuracy is 85.17%, some of which are misjudged as arable land, mainly because it is difficult to distinguish paddy fields and coastal farmland in single-phase remote sensing image data. In contrast, the user accuracy of mangroves is 100% and the producer accuracy is 36.84%. Figure 3 (left) shows the key area of mangrove distribution in the study area, Qi’ao Island. It can be seen that some mangrove areas on the west side were missed.

### 3.2. Integrated Time Series NDVI Data Classification

The spatiotemporal distribution of NDVI is shown in Figure 4. These results show that the vegetation coverage of forest and mangrove areas is significantly higher than those of arable land and grassland, and the latter two types can themselves be distinguished by changes in time series characteristics. The vegetation coverage of arable land and grassland is generally similar, but a clear distinction between the two can be identified in April (2H) and September (2H).

The integrated time series NDVI data classification results are shown in Figure 5 and Table 4. The two texture features (VV_dvar and VH_diss) are still the most important. In the time series NDVI data, the most important feature is October (1H), followed by September (2H). The classification results show that the identification of arable land and grassland has become significantly more accurate, especially in the northeastern corner of the study area. There has been a significant reduction in wrongly classified grassland types, and total accuracy has increased by 3.33%, reaching a level of 80.95%. In addition, the user accuracy and producer accuracy of the water both reached 100%, which shows that the time series NDVI data can better distinguish paddy fields and coastal farmland from water. The user accuracy of arable land increased from the previous 65.00% to 85.71%, and the improvement was the largest, which indicated that the time series NDVI data had a strong ability to identify vegetation with obvious seasonality, such as arable land.

### 3.3. Integrated Time Series SAR Data Classification

The spatiotemporal distribution of SAR data is shown in Figure 6. The results show that VH polarimetric SAR data only weakly distinguish forests, mangroves, and impermeable layers from other types, making misclassification more likely, while the VV polarimetric SAR data strongly distinguish the three types. For this reason, in this study, we used VV polarimetric SAR data for the time series SAR images and integrated the optical remote sensing data to classify land cover types in the study area.

The integrated time series SAR data classification results are shown in Figure 7 and Table 5. In the optical remote sensing data, texture features are of the greatest importance, with EVI being the most important of all. With the time series SAR data, October (2H) and September (2H) are the most important. The classification results show improved accuracy in the identification of mangrove, arable land, and grassland, with reduced misclassification between these three types and forests. Overall, accuracy has improved significantly. Compared with the classification results of the single-temporal data, total accuracy has increased by 4.76% to 82.38%. In addition, the classification results obtained from the time series SAR data are more accurate than those obtained from the time series NDVI data. In addition, the producer accuracy of mangroves increased from 36.84% to 63.16%, and the improvement effect was the most obvious, which indicated that time series SAR data could better identify mangroves and forests and other easily mixed vegetation types.

### 3.4. Integrated Active and Passive Time Series Data Classification

All data were integrated, including the time series NDVI data, time series SAR data, and optical remote sensing data to obtain a final classification of land cover types, and the results are shown in Figure 8. Regarding feature importance, NRWI, VH_diss, and B2 are the top three in all features, followed by the April (1H) and October (2H) features in the time series NDVI data, while the most important feature of the SAR data is September (1H). The classification results (Table 6) show some improved precision in identifying vegetation types, such as forest, mangrove, and arable land, with significantly increased accuracy for grassland. The overall level of accuracy is 84.29%, which is 6.67% higher than the single-temporal remote sensing classification. This is a finding of potentially great importance for the identification of land cover types in urban agglomerations with complex ecosystems in subtropical areas dominated by evergreen broad-leaved forests. The integrated active and passive time series data can better distinguish arable land, mangroves, forests, grasslands, and other vegetation types. Their accuracy has been improved to varying degrees, and the amplitude is about 10%. In particular, the accuracy of grass has the highest increase, reaching 21.88%.

## 4. Discussion

This study focused on analyzing the effect of active and passive time series data on the extraction of land cover types in subtropical regions. Our results show that active and passive remote sensing data present different image characteristics at different periods during the year. Therefore, some factors not considered in this study may have a certain impact on the results. Here are the two main types of analysis: window size on extraction of texture feature information and the classification method.

### 4.1. Effect of Window Size on Extraction of Texture Feature Information

A total of 36 feature sets were formed based on the VV polarization mode and VH polarization mode of SAR data in this study, and a 7 × 7 window was selected, according to the size and distribution of the overall ground object types in the study area. Figure 9 shows the true color image, false color image of Qi’ao Island, and texture feature images under different windows. Texture features include variance, dissimilarity, and entropy of the gray level co-occurrence matrix, while windows include 3 × 3, 5 × 5, 7 × 7, 9 × 9. Different texture feature windows have different recognition capabilities, and there is a most suitable feature extraction window for different land cover types [59]. For example, the internal structure of forest types is complex and heterogeneity is large, and a larger texture feature window makes it easier to extract forest types. Future studies may analyze the texture features formed by other windows and select characteristic texture feature windows for different types of ground cover to further improve classification accuracy. Additionally, more texture feature datasets might be extracted, based on each wavelength band and vegetation index of optical remote sensing data. To use all texture feature information efficiently and reasonably, more targeted feature extraction methods may be required.

### 4.2. Importance Analysis of Different Vegetation Indexes

The vegetation information captured by different spectral channels has some degree of correlation with vegetation type, growth status, health, and so on. However, single-band data analysis has obvious limitations in distinguishing different vegetation types for complex ecosystem urban agglomeration in tropical and subtropical regions. Therefore, we often choose to generate indicative vegetation indices from multispectral remote sensing data through analytical operations (linear or non-linear combinations, such as addition, subtraction, multiplication, division). In this study, the input of different feature sets not only has an effect on the identification accuracy, but also changes the importance of different vegetation indices. We analyzed the importance characteristics of different vegetation indices in four scenarios, as shown in Figure 10. As the number of feature sets increases, the importance of each feature decreases relatively, so in Figure 10, we only need to compare the relative importance of vegetation index in each case. From the analysis, we can see that EVI is very important in the classification of single-temporal remote sensing data and time series SAR data as feature sets, far exceeding other vegetation indices. However, when the time series NDVI data are input into the feature set, the importance of EVI decreases significantly. This is because NDVI itself is also a vegetation index, and its input reduces the role of other vegetation indices in classification. Other vegetation indices do not differ much, except that when integrating active and passive time series data, the NRWI index is the most important, exceeding 15%. Through comprehensive analysis, it is found that each vegetation index is generally for one land cover type. The main function of EVI is to extract vegetation. Due to the complexity of vegetation types in GBA, including forests, grasslands, mangroves, and arable land, the EVI can greatly improve the ability to distinguish different vegetation types. The function of NDWI is to extract water body, and RGRI can better remove the interference of soil background and is more conducive to extracting information of pervious layer. The main function of NDBI is to extract impervious layers, such as buildings. For different vegetation indices in different regions, the impact of different feature sets needs to be further studied.

### 4.3. Comparative Analysis of Different Classification Methods

As the study method, the random forest classifier is mainly used because the classifier itself contains multiple classifiers, and each decision tree is a classifier, so it integrates the advantages of multiple classifiers. This can better improve the classification ability and classification accuracy [57,60]. Currently, there are many methods to classify land cover, such as computer-based deep learning techniques for a variety of parametric, non-parametric, and supervised and unsupervised classification methods, including support vector machines, neural networks, maximum likelihood classification, minimum distance classification, decision tree classification, spectral mixing analysis, spectral information divergence, and the spectral angle mapper [61]. Figure 11 shows the classification results of Qi’ao Island using the maximum likelihood method and the support vector machine method. The maximum likelihood method found it difficult to distinguish between mangroves and forests, and the mixture of the two is more serious. Although the support vector machine method can distinguish between forests and mangroves to a certain extent, the recognition rate of the two is still low. In addition, these two methods also have a mixed phenomenon among vegetation types, such as forest, grassland, and arable land, which will greatly affect the accuracy of classification and recognition. From the perspective of the entire study area, how different classification methods differ for different land cover types requires further in-depth study in the future.

## 5. Conclusions

In this study, we took the GBA as a study area and Sentinel-1 and Sentinel-2 data as the active and passive remote sensing time series data to distinguish typical ecosystem types in subtropical urban agglomerations. The results show the following: (1) The importance of different features varies widely in different types of ecosystems. For grassland and arable land, two specific texture features (VV_dvar, and VH_diss) are most important. In forest and mangrove areas, SAR data for the months of October and September are most important. (2) Both time series NDVI data and time series SAR data significantly improved classification accuracy, by 3.33% and 4.76%, to levels of 80.95% and 82.38%, respectively. When the active and passive time series data were simultaneously integrated, accuracy was further improved, to 84.29%. (3) Time series passive data (NDVI) serve best to distinguish grassland from arable land, while time series active data (SAR data) are best able to distinguish mangrove from forest. The integration of active and passive time series data also improves precision in distinguishing vegetation ecosystem types, such as forest, mangrove, arable land, and, especially, grassland, where we found an accuracy improvement of 21.88%. These results show that the remote sensing image classification model integrating active and passive time series data can achieve accurate and rapid identification of land cover types in the complex ecosystems of urban agglomerations in subtropical regions and provide important information support for decision-makers in urban agglomerations.

## Figures and Tables

**Figure 1 ijerph-19-15108-f001:**
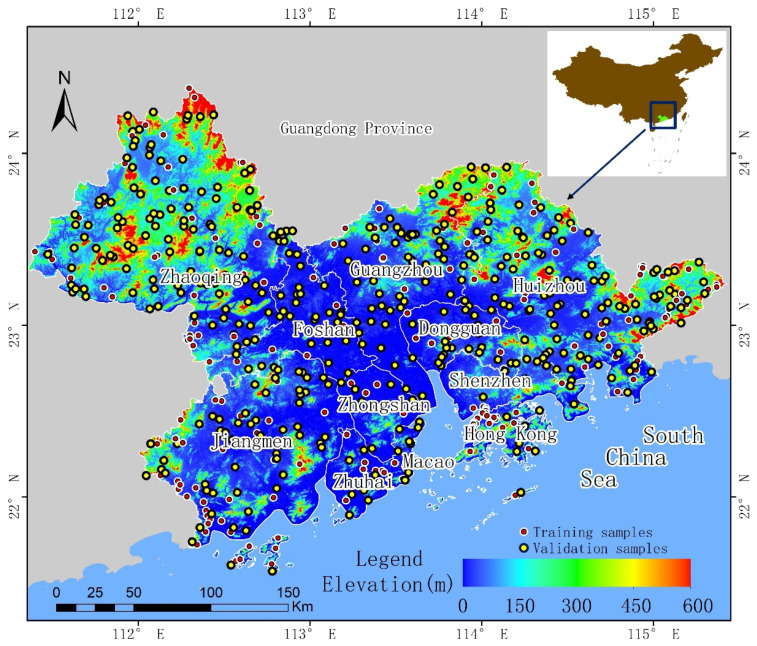
Location of the study area, GBA, China. It includes nine cities and two special administrative regions, namely, Guangzhou, Shenzhen, Zhuhai, Foshan, Huizhou, Dongguan, Zhongshan, Jiangmen, and Zhaoqing; Hong Kong Special Administrative Region and Macao Special Administrative Region.

**Figure 2 ijerph-19-15108-f002:**
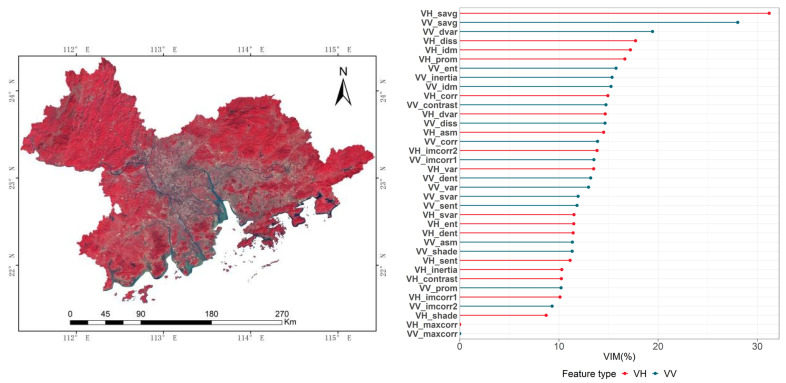
Single-temporal remote sensing image feature analysis. (**Left**): Sentinel-2 false-color composite image (red—B8; green—B4; blue—B2). (**Right**): Texture feature importance analysis.

**Figure 3 ijerph-19-15108-f003:**
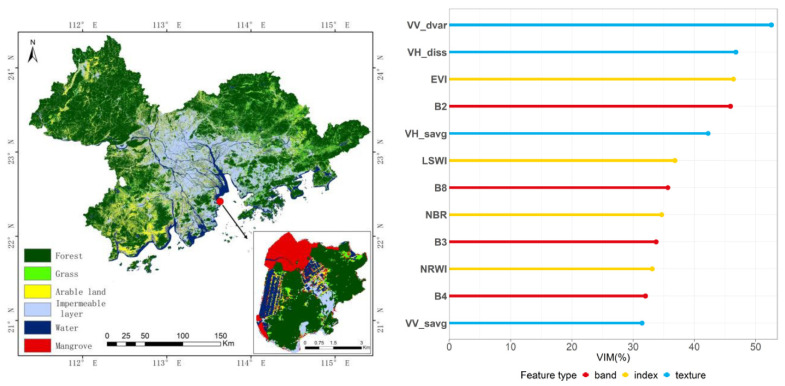
Single-temporal image classification results. (**Left**): Spatial distribution of land cover types by the single-temporal images classification. (**Right**): Classification feature importance analysis.

**Figure 4 ijerph-19-15108-f004:**
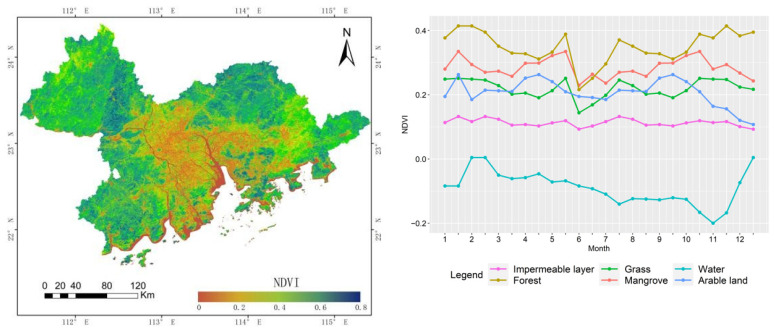
Spatiotemporal distribution of NDVI. (**Left**): Spatial distribution of the annual maximum NDVI. (**Right**): Annual NDVI variation curves for land cover types after Savizky–Golay filtering.

**Figure 5 ijerph-19-15108-f005:**
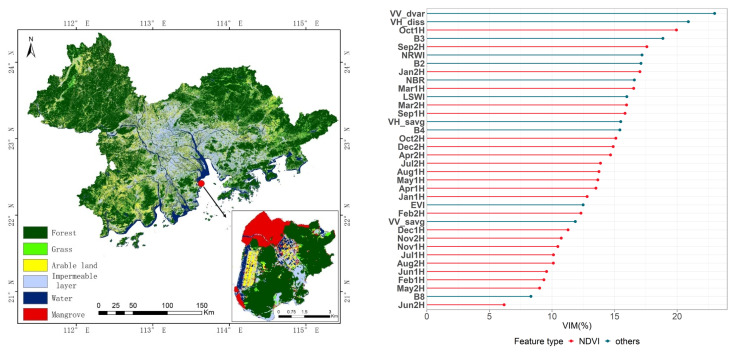
Integrated time series NDVI data classification results. (**Left**): Spatial distribution of land cover types by integrated time series NDVI data classification. (**Right**): Classification feature importance analysis.

**Figure 6 ijerph-19-15108-f006:**
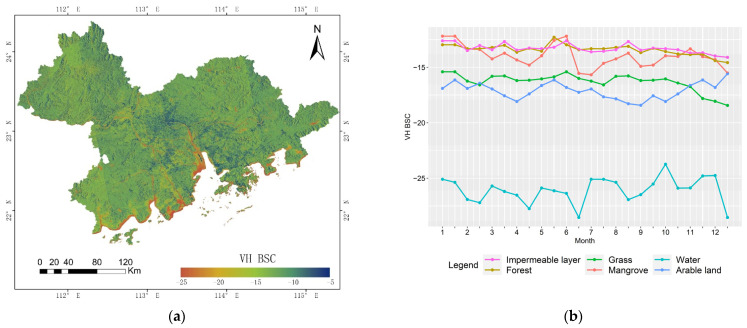
Spatiotemporal distribution of SAR data. (**a**) Spatial distribution of the VH polarimetric backscattering coefficient (VH BSC) for July. (**b**) VH BSC variation curves for land cover types. (**c**) Spatial distribution of the VV polarimetric backscattering coefficient (VV BSC) for July. (**d**) VV BSC variation curves for land cover types.

**Figure 7 ijerph-19-15108-f007:**
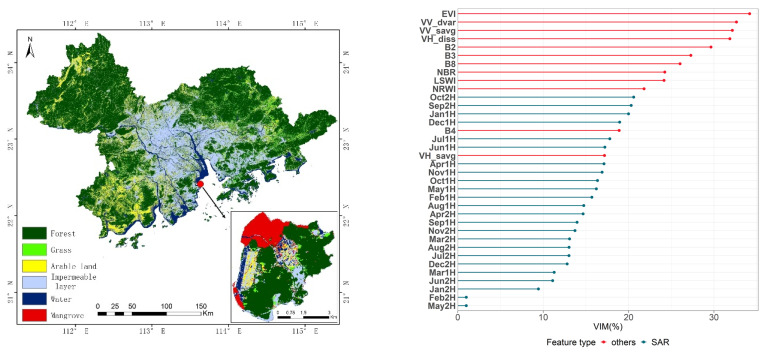
Integrated time series SAR data classification results. (**Left**): Spatial distribution of land cover types by integrated time series SAR data classification. (**Right**): Classification feature importance analysis.

**Figure 8 ijerph-19-15108-f008:**
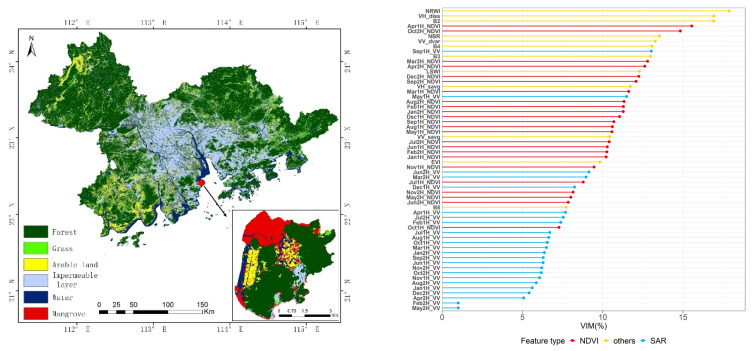
Integrated active and passive time series data classification results. (**Left**): Spatial distribution of land cover types by integrated time series SAR data classification. (**Right**): Classification feature importance analysis.

**Figure 9 ijerph-19-15108-f009:**
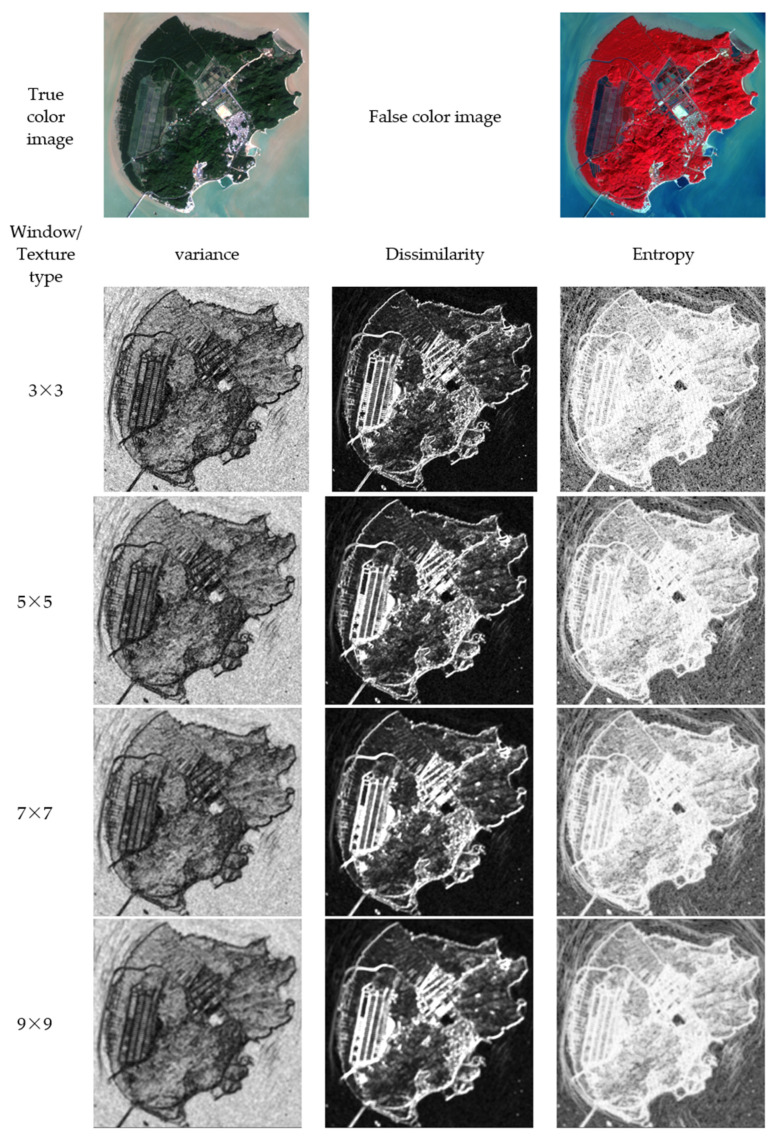
True color image, false color image of Qi’ao Island, and texture feature images under different windows. Texture Features: variance, dissimilarity, and entropy of the gray level co-occurrence matrix. Windows: 3 × 3, 5 × 5, 7 × 7, 9 × 9.

**Figure 10 ijerph-19-15108-f010:**
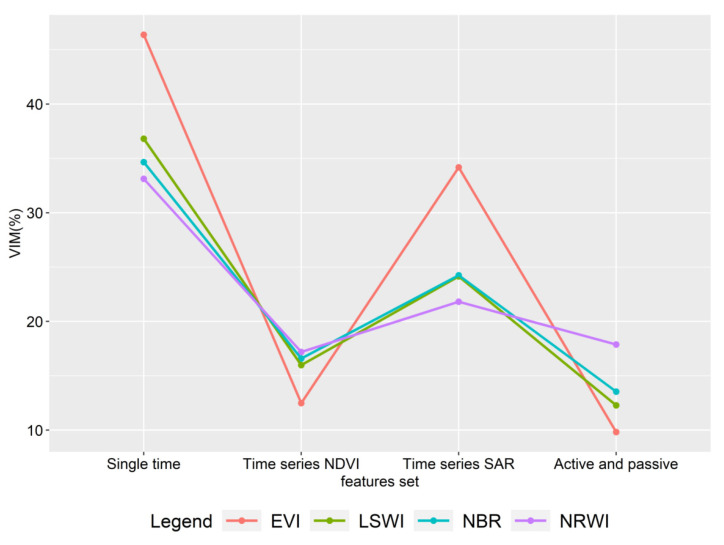
Importance characteristics of different vegetation indices in four scenarios.

**Figure 11 ijerph-19-15108-f011:**
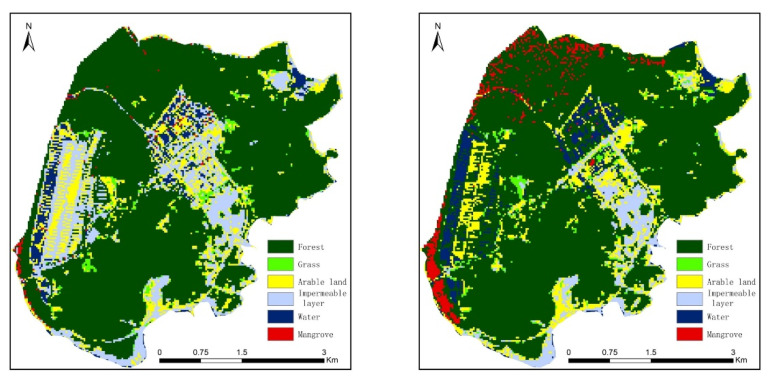
Land cover classification results of other classification methods (taking Qi’ao Island as an example): maximum likelihood method (**Left**), support vector machine method (**Right**).

**Table 1 ijerph-19-15108-t001:** Land cover classification system and definitions.

No.	Type	Definition
1	Forest	Forest land dominated by trees, with canopy closure ≥ 0.2
2	Grass	Land that produces herbaceous plants
3	Arable land	Land where crops are the main surface type
4	Impermeable layer	Artificial surfaces such as buildings, roads, factories, etc.
5	Mangrove	Wetland woody plant communities composed of evergreen trees or shrubs dominated by mangroves
6	Water	Inland waters, beaches, ditches, swamps, hydraulic structures, etc.

**Table 2 ijerph-19-15108-t002:** Features set.

Sensor	Feature Type	Feature Variable
Sentinel-1	Polarization mode	VH
		VV
	Texture features	Fourteen GLCM features proposed by Haralick, and four additional features from Conners
Sentinel-2	Spectral features	Blue band (B2)
		Green band (B3)
		Red band (B4)
		Near-infrared band (NIR, B8)
	Index features	Normalized difference vegetation index, NDVI
		Enhanced vegetation index, EVI
		Normalized difference water index, NDWI
		Red–green ratio index, RGRI
		Normalized difference built-up index, NDBI

**Table 3 ijerph-19-15108-t003:** Confusion matrix of single-temporal image classification results.

Types	Water	Arable Land	Impermeable Layer	Mangrove	Forest	Grass	Sum	Producer Accuracy %
Water	42	0	0	0	0	0	42	100.00
Arable land	7	39	11	0	11	12	80	48.75
Impermeable layer	0	4	67	0	0	0	71	94.37
Mangrove	0	4	0	7	4	4	19	36.84
Forest	0	0	0	0	140	10	150	93.33
Grass	0	13	0	0	14	31	58	53.45
Sum	49	60	78	7	169	57	420	
User accuracy %	85.71	65.00	85.90	100.00	82.84	54.39		77.62
Kappa coefficient	0.7080							

**Table 4 ijerph-19-15108-t004:** Confusion matrix of classification results of integrated time series NDVI data.

Types	Water	Arable Land	Impermeable Layer	Mangrove	Forest	Grass	Sum	Producer Accuracy %
Water	42	0	0	0	0	0	42	100.00
Arable land	0	48	18	0	10	4	80	60.00
Impermeable layer	0	4	67	0	0	0	71	94.37
Mangrove	0	4	0	7	4	4	19	36.84
Forest	0	0	0	0	139	11	150	92.67
Grass	0	0	11	0	10	37	58	63.79
Sum	42	56	96	7	163	56	420	
User accuracy %	100.00	85.71	69.79	100.00	85.28	66.07		80.95
Kappa coefficient	0.7520							

**Table 5 ijerph-19-15108-t005:** Confusion matrix of classification results of integrated time series SAR data.

Types	Water	Arable Land	Impermeable Layer	Mangrove	Forest	Grass	Sum	Producer Accuracy %
Water	42	0	0	0	0	0	42	100.00
Arable land	7	41	18	0	7	7	80	51.25
Impermeable layer	0	4	67	0	0	0	71	94.37
Mangrove	0	0	0	12	7	0	19	63.16
Forest	0	0	0	0	143	7	150	95.33
Grass	0	6	0	0	11	41	58	70.69
Sum	49	51	85	12	168	55	420	
User accuracy %	85.71	80.39	78.82	100.00	85.12	74.55		82.38
Kappa coefficient	0.7708							

**Table 6 ijerph-19-15108-t006:** Confusion matrix of results of integrated active and passive time series data.

Types	Water	Arable Land	Impermeable Layer	Mangrove	Forest	Grass	Sum	Producer Accuracy %
Water	42	0	0	0	0	0	42	100.00
Arable land	7	46	10	0	10	7	80	57.50
Impermeable layer	0	5	66	0	0	0	71	92.96
Mangrove	0	0	0	12	7	0	19	63.16
Forest	0	0	0	0	143	7	150	95.33
Grass	0	7	0	0	6	45	58	77.59
Sum	49	58	76	12	166	59	420	
User accuracy %	85.71	79.31	86.84	100.00	86.14	76.27		84.29
Kappa coefficient	0.7958							

## Data Availability

All data has been presented in the paper, and no new data were created or analyzed in this study. Data sharing is not applicable to this article.

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
