# Peer review of "Identification of Typical Ecosystem Types by Integrating Active and Passive Time Series Data of the Guangdong–Hong Kong–Macao Greater Bay Area, China"

_ijerph, 2022, doi:10.3390/ijerph192215108_

Round 1

Reviewer 1 Report

This is a very interesting and data-rich manuscript about the use of remote sensing data to distinguish ecosystem types in subtropical urban agglomerations and overcome weaknesses in ecosystem identification. Manuscript is relatively well-written with a large amount of data and good figures. Methods are well-explained. Results are strong. Discussion lacks of a more consistent focus on the role of your data in environmental projects (authors rightly evidenced as these data are usful to support decisions).

In abstract, the following sentence is good: ‘The integration of active and passive time series data also improves precision in distinguishing vegetation ecosystem types such as forest, mangrove, arable land, and, especially, grassland, where we found an accuracy improvement of 21.88%. This study provides important information support for decision makers in urban agglomerations, and for researchers of ecological effects.’ However, the first section of sentence is very specific about the techniques used, the second section is too general about possible implications of your data in decision making. An intermediate sentence explaining the role of your data in projects of landscape planning, ecological network planning, ecological restoration and so on, should be added. Some further sentences about the decision making as a strategic step inside the project cycle should be added (see below).

Row 36-47. Ok, there are many changes at landscape level but I would read some implications of these changes on ecosystems. For example, Authors should add some outcomes on biodiversity, ecosystems and landscapes. What about landscape fragmentations (see Lindenmayer, Nix, Crooks, Fahrig and others; e.g.  Forest Ecology and Management159(3), 203-216.).

Authors stated that your data may support decision makers in abstract and introduction. They reported the concept of ‘decision tree’ (e.g. in row 279). Ok. But I would like to read something more structured about the role these decision trees and decision support systems within environmental projects. In this regard, I suggest to locate your logic inside the project cycle applied to environmental/conservation arenas. See, for example, Journal for Nature Conservation41, 63-72 about project cycle (and all the steps, included the decision phase). The role of your data in the logic of project cycle could be improved adding some sentences in this regard. I think that the ms could could also increase the number of readers.

References are too Chinese-focused. This topic and the related techniques are worldwide known and some further seminal references not only focused on East Asia should be included. For example, see: Roy, P. S., & Tomar, S. (2000). Biodiversity characterization at landscape level using geospatial modelling technique. Biological conservation95(1), 95-109, and Newbold, T., Hudson, L. N., Phillips, H. R., Hill, S. L., Contu, S., Lysenko, I., ... & Purvis, A. (2014). A global model of the response of tropical and sub-tropical forest biodiversity to anthropogenic pressures. Proceedings of the Royal Society B: Biological Sciences281(1792), 20141371.

Add the role of anonymous reviewers in the acknowledgments.

I would like to read a revised version of this good ms.

Have a nice work.

Author Response

We greatly appreciate the reviewer’s comments on our manuscript. We have revised our manuscript according to these comments. The changes are shown in the revised manuscript.  The attachment is the point-to-point response addressing these comments.

Reviewer 2 Report

see attached file

Reviewer 3 Report

I am glad to receive the review invitation for the article titled Identification of Typical Ecosystem Types by Integrating Active and Passive Time Series Data of the Guangdong–Hong 3 Kong–Macao Greater Bay Area, China from Int. J. Environ. Res. Public Health. I appreciate author’s intension to choose an insightful study, particularly one that focuses on land cover classification (ecosystem) in a mix environment where distinct boundary between individual land covers are complex to differentiate. After careful reading, I'd say the article is well drafted (but English could have been more improved), hence suitable for publication but needs a bit improvement. Having said that, I have recommendations that I believe the authors will find useful and that, if implemented, would make this research suitable for publication in the journal.

 (1)Title

Title of the article is somewhat unclear as the term typical ecosystem types made the title more complex, what I see is the manuscript was focused on land cover or  I can say ecosystem mapping and offer an techniques which could provide the best classification outcomes. Consider revising the title. I wonder why this research is not land cover classification, need a rationale somewhere in the manuscript. You can refer the article where authors demonstrated different between land cover and ecosystem in there study (https://www.sciencedirect.com/science/article/pii/S221242091931578X).

 (2) Abstract

Apparently the given abstract is ok but authors need to slightly modify the abstract. The study's implication is unclear; how does the present approach can help other geospatial analysis for example better change detection etc.

 (3) Introduction

 The introduction section is not written as well as it should be for a scientific article; in my opinion, the authors jump from one topic to another without a connecting sentence. For instance, line 38 to 48. In the Introduction, it is necessary to reconstruct the storyline of this manuscript. The introduction is the opening part of the scientific story to attract an audience and suggest the direction of your research. For this reason, you need to identify the problem that drives the research and introduce the key characters. If then, using the key characters, it is required to intertwine the scientific story concisely, systematically, and logically. I have few comments which might be beneficial if included in the revised manuscript. In this study, the key characters are Guangdong–Hong Kong–Macao Greater Bay Area (GBA); typical ecosystem types; active and passive data; time series data. What I can see from the First paragraph of the article is the generic demonstration on china’s urbanization and land cover changes, but authors put a specific region i.e., Guangdong–Hong Kong–Macao Greater Bay Area (GBA)- this is an example, you might have all the relevant information in the introduction section but not an organized way. Please reorder the keywords or rearrange the paragraphs in introduction section. Last but not least, if there is no problem with the length of the article, I would suggest that the author somewhat enhance the introduction section because, in the format as it is now, it appears that authors quickly move from one section to another. Other comments are given below:

   Line 65-67 is more likely a study area description or use them as a rationale of study area selection. Need a proper reference in line 36-38. Need a proper reference in line 83-84; 85-87.  Please rewrite lines 88-95

 (4) Materials and Methods

# A method overview is strongly desirable because it can give readers a better sense of the analytical techniques used in this work.

# Overview of the study area – why this particular area is taken as a study area, need an explicit rationale. Need references in this section where statistics and other relevant information’s are given relating to the study area (line 100-104;  Please change the color of South China Sea area to blue and other land parts outside of the study area to write or grey. Please reduce the size of the circles representing sampling and validation. 

# not clear ( line 126-127) please provide date of google earth data and I am certain that not every part of the study area is covered by google earth high resolution data, how did the authors resolve the problem during collection validation points if any portion of the study area was missing.

# Line 129- 130, code of the analysis is highly desirable to submit as supplementary data so that readers will be benefited

The validation data were randomly generated at 120 sample points, and then manually interpreted based on Google Earth” a supplementary is highly desirable for the result of manual validation

(5) Result

All the figures and graphs need to be improved. I would recommend to enlarge the figures and also make the graph scale more legible.  A bit more elaboration on accuracy assessment in each section is recommend for a better understanding of active and passive remote sensing data

(6) Discussion

In the discussion section I do not see much cross-references based on the result of the study, it is highly desirable a detail discussion on the results of the study and supplement them with other studies. If possible please elaborate limitation, future direction and policy implication of this study. Also authors need to discuss about this current classification scheme and its suitability over other classification like object based image analysis – which is considered to be one of the best techniques in land cover classification.

(7) Conclusion

 Generally ok but need a bit elaborate the results in this section. The line “for researchers of ecological effects” please provide some evidences.

Overall, write-up across all the sections of this manuscript need to be improved. Please avoid using the term WE always. I would recommend to proof the manuscript with an academic English editor.

Reviewer 4 Report

The authors present an article where they try to classify and analyse the effectiveness of time series SAR data and time series NDVI data in the identification of land cover types in the Guangdong-Hong Kong-Macao Greater Bay Area, China, the article combines active and passive remote sensing data (Sentinel 1 and 2) and uses machine learning algorithms. They seek to create a faster and more precise technique for classifying the many forms of land cover in the intricate ecosystems of subtropical urban agglomerations aiming to offer crucial information assistance for ecological researchers and urban agglomeration decision-makers.

The study is interesting and fits within the scope of the International Journal of Environmental Research and Public Health. However, the manuscript has many flaws and inconsistencies that prevent its publication in the journal.

The abstract contains conclusions that are not reflected in the conclusions section or the discussion section. For example, concerning line 25: "(3) Time series passive data (NDVI) serve best to distinguish grassland from arable land, while time series active data (SAR data) are best able to distinguish mangrove from forest", the authors only indicate related information in the results. But this information is neither discussed nor presented in the conclusions. Therefore, the authors have to keep the different sections consistent.

The introduction section needs further explanations. There are inconsistencies between what the authors state in the text and the references they provide. Moreover, the authors state that "commonly used ecosystem identification methods have always lacked accuracy in the identification methods in subtropical urban agglomeration", but they don't link this statement with the specified case of study: the Guangdong–Hong Kong–Macao Greater Bay Area. Besides, they don't properly explain these "common" methods. Given the importance of these methods for the article, the authors should create a paragraph in the introduction to explain it and link it to the discussion (some parts of the discussion could be moved for this purpose). In addition, the results are clear, but the authors mixed much information and therefore it is not clear to this reviewer what scientific gap they are trying to fill.

There is redundant information and content in the material and methods section, that could not be necessary for the article, making it too verbose. For example, the authors talk about different vegetation indices in material and methods but do not discuss them in the results or discussion.

The results section is well presented, the figures look good and useful, and the captions are accurate.

Finally, the discussion and conclusions sections are extremely weak, especially the discussion. The discussion sounds like an introduction in several parts; therefore, these parts should be moved. Further discussion of the results obtained is needed, and, in fact, some just repeat results. For instance, the authors present sound results in line 249 (total accuracy has increased by 4.76% to 82.38%) but fail to link them to the discussion. There is a lack of further discussion about the suitability of the methods (machine learning…) used in this work. The results show user accuracy and Kappa coefficients, but they don't discuss anything. Moreover, the authors discuss several times what can be done in future studies (lines 286-294-300), but do not discuss what has been done in this one, and no comparison with results from other articles has been made. At least a subsection in the discussion comparing with other methods should be developed to support this statement.

Finally, a comprehensive English language revision is needed all over the manuscript.

Therefore, I recommend the rejection of this manuscript because it does not reach the high-quality standards for being published in the International Journal of Environmental Research and Public Health.

Specific comments:

Line 17

Please specify "China".

Line 19

"we combined active and passive remote sensing data with time series data"

Sentinel 1 and 2 are the active and passive sensors. What is the additional "time series data" that you combined? Or is it time series data from sentinel 1 and 2? Please, clarify.

Line 22

Authors should indicate the meaning of acronyms the first time they appear (SAR).

Line 23

"The use of active and passive remote sensing data"

Separately or combined? I suppose it is separated, but please specify.

Line 44-45

It is not clear to this reviewer what means "single-temporal remote sensing data". Do the authors refer to time series generated using data from one sensor/satellite?

The authors support "single-temporal remote sensing data fail to fully describe land cover" referencing article [3], but it is unclear how that article states this.

Line 66

"In such regions, it is difficult to accurately…" Is it really more difficult to distinguish vegetation in subtropical regions than in other regions? The authors should elaborate on this point and include references.

Line 68

"The use of time series optical data is more likely to produce accurate results"

More likely to produce accurate results? Compared to what? Other information from other satellites? The authors just indicated in the previous paragraph that the combination of optical data with different types of information leads to an improvement in classification accuracy.

Moreover, reference [14] states that Google Earth high resolution image data was used for field confirmation, so it is unclear what the link between the reference and the provided information is.

Line 92

"quicker and more accurate method of identifying land cover types in the complex ecosystems of subtropical urban agglomerations"

However, no comparison with papers from other articles and no discussion has been made. Therefore, at least a subsection in the discussion comparing with other methods should be developed to support this statement.

line 122-124

"We obtained time series NDVI data by calculating the optimal value of the image after the first and second halves (1H, 2H) of each month (a 124 total of 24 scenes)"

It is not clear. Please, develop this point.

Line 126

"manual interpretation"

I understand some processes must be carried out manually. However, the authors should explain the criteria they followed.

Line 138-145

The authors estate that they studied time series NDVI (line 91). However, they calculate here several vegetation indices.

Line 176-186

Is it necessary for the comprehension of the paper the paragraph regarding "2.4.2. RF Classification Algorithm"? the section is not linked to the paper. It is a generic explanation.

Moreover, why the authors employed random forest instead of other methodologies?

Line 177

"We used a random forest algorithm [that] is based on a single decision tree"

Correct English

Line 182

Models were instead of “Models was”

Correct English

Line 184

"the out-of-bag error was used", but no metrics about OOB are in the results and discussion section.

Line 206

grasslands instead of “grass landsCorrect English

line 246

“[the] most important” Correct English

Line 274-285

This part should be moved to the introduction section.

Line 285-286

"we used only the random forest classifier, which may have certain deficiencies in information extraction" what are the results that support this statement? Why did the authors select this machine learning method if authors believe it?

Line 287

"excavate" I don't understand the meaning in this context.

Line 289-292

This part should be moved to the introduction section.

Line 302

"This study focused on analysing the effect of active and passive time series data on the extraction of land cover types in subtropical regions."

However, no discussion has been presented.

Line 315

Move "respectively" to "of 80.95% and 82.38%, [respectively]"

Round 2

Reviewer 3 Report

I'm happy with the corrections made the authors 

Author Response

We deeply appreciate your consideration of our manuscript.

Reviewer 4 Report

The manuscript entitled “Identification of Typical Ecosystem Types by Integrating Active and Passive Time Series Data of the Guangdong–Hong Kong–Macao Greater Bay Area, China”, authored by Changlong Li, Yan Wang, Zhihai Gao, Bin Sun, He Xing and Yu Zang is a re-submission from a former manuscript that I had the opportunity to review. In summary, the authors combine in this work active and passive remote sensing data (Sentinel 1 and 2) and they use machine learning algorithms. They seek to create a faster and more precise technique for classifying the many forms of land cover in the intricate ecosystems of subtropical urban agglomerations aiming to offer crucial information assistance for ecological researchers and urban agglomeration decision-makers.

The study is interesting and fits within the scope of the International Journal of Environmental Research and Public Health. The authors have greatly improved the manuscript and have responded to almost all questions. However, there are still some comments that they did not address adequately and, therefore, some portions of the text should be rephrased:

The main concern is that for this reviewer it is not yet clear what is the contribution of each variable for the classification, specifically vegetation indices (VI). I believe at least a discussion about VIs should be addressed (because of their importance in remote sensing). Authors state that “The different vegetation indices in material and methods were used as classification features in single-temporal image classification”, however they do not discuss key aspects such as the implications, or the importance of each VI in the classification.

The authors performed a classification feature importance analysis and plotted the information on several figures (2,3,5,7,8), but they only indicate in the text the most important variables per figure, that is:

“the effects of EVI and B2 are similar and rank third and fourth, respectively, in the feature importance scores” (line 272)

“In the optical remote sensing data, texture features are of the greatest importance, 325 with EVI is the most important of all” (line 326)

And

“Regarding feature importance, NRWI, VH_diss, and B2 are the top three in all features” (line 343)

However, they don’t discuss these results. These analyses are critical to understanding this study and authors should elaborate more on the rationale behind these figures. Therefore, a discussion about the contribution of each variable or, at least, each vegetation index should be addressed in the discussion section. (including why it achieved its value). For example, why EVI or NDWI achieved better results than other vegetation indices?

I recommend a minor revision of this manuscript prior to its acceptance for being published in the International Journal of Environmental Research and Public Health.

Specific comments:

Figure 4 contains Chinese characters.

Although the authors state an English revision was conducted all over the manuscript, I believe that the English language should be revised again to correct some minor errors that still remain in the manuscript. Examples:

Line 397

“The maximum likelihood method is difficult to distinguish between mangroves and forests, and the mixture of the two is more serious”

The verb in this cause should be “found it difficult” instead of “is difficult”

Line 387

“As the study method, the random forest classifier is mainly [XXX] because the classifier itself contains multiple classifiers”

Is mainly… what? employed? Used?
